# Analysis of bacterial vaginosis, the vaginal microbiome, and sexually transmitted infections following the provision of menstrual cups in Kenyan schools: Results of a nested study within a cluster randomized controlled trial

**Supriya D. Mehta**[1,2]\*, **Garazi Zulaika**[3], **Walter Agingu**[4], **Elizabeth Nyothach**[5], **Runa Bhaumik**[2], **Stefan J. Green**[6], **Anna Maria van Eijk**[3], **Daniel Kwaro**[5], **Fredrick Otieno**[4], **Penelope Phillips-Howard**[3]

**1** Division of Infectious Disease Medicine, Rush University College of Medicine, Chicago, Illinois, United States of America, **2** Division of Epidemiology & Biostatistics, University of Illinois Chicago School of Public Health, Chicago, Illinois, United States of America, **3** Department of Clinical Sciences, Liverpool School of Tropical Medicine, Liverpool, United Kingdom, **4** Nyanza Reproductive Health Society, Kisumu, Kenya, **5** Centre for Global Health Research, Kenya Medical Research Institute, Kisumu, Kenya, **6** Department of Internal Medicine and Genomics and Microbiome Core Facility, Rush University, Chicago, Illinois, United States of America

\* Supriyad@uic.edu

## Abstract

### Background

Nonhygienic products for managing menstruation are reported to cause reproductive tract infections. Menstrual cups are a potential solution. We assessed whether menstrual cups would reduce bacterial vaginosis (BV), vaginal microbiome (VMB), and sexually transmitted infections (STIs) as studies have not evaluated this.

### Methods and findings

A cluster randomized controlled trial was performed in 96 Kenyan secondary schools, randomized (1:1:1:1) to control, menstrual cup, cash transfer, or menstrual cup plus cash transfer. This substudy assessing the impact of menstrual cups on BV, VMB, and STIs, included 6 schools from the control (3) and menstrual cup only (3) groups, both receiving BV and STI testing and treatment at each visit. Self-collected vaginal swabs were used to measure VMB (16S rRNA gene amplicon sequencing), BV (Nugent score), and STIs. STIs were a composite of *Chlamydia trachomatis* and *Neisseria gonorrhoeae* (nucleic acid amplification test) and *Trichomonas vaginalis* (rapid immunochromatographic assay). Participants were not masked and were followed for 30 months. The primary outcome was diagnosis of BV; secondary outcomes were VMB and STIs. Intention-to-treat blinded analyses used mixed effects generalized linear regressions, with random effects term for school. The study was

**Data Availability Statement:** Raw sequence data (FASTQ files) were deposited in the National Center for Biotechnology Information (NCBI) Sequence Read Archive (SRA), under BioProject identifier PRJNA746243. This study was conducted with approval from the Kenya Medical Research Institute (KEMRI) Scientific and Ethics Review Unit (SERU), which requires that data be released from any KEMRI-based Kenyan studies (including de-identified data) only after their written approval for additional analyses. In accordance, data for this study will be available upon request, after obtaining written approval for the proposed analysis from the KEMRI SERU. Their application forms and guidelines can be accessed at https://www.kemri.org/seru-overview. To request these data, please contact the KEMRI SERU at seru@kemri.org.

**Funding:** This study was supported by the National Institutes of Health Eunice Shriver National Institute of Child Health and Human Development (R01-HD093780 to SDM), and the Joint Global Health Trials Initiative (UK-Medical Research Council/Department for International Development/Wellcome Trust/Department of Health and Social Care; MR/N006046/1 to PPH). The funders had no role in the design of the study, the collection, analysis, and interpretation of data, or in writing the manuscript.

**Competing interests:** The authors have declared that no competing interests exist.

**Abbreviations:** BIOM, biological observation matrix; BV, bacterial vaginosis; CaCHe, Cups and Community Health; CCT, conditional cash transfer; COVID-19, Coronavirus Disease 2019; CST, community state type; CT, *Chlamydia trachomatis*; GLMM, generalized linear mixed model; GRC, Genome Research Core; HDSS, health and demographic surveillance system; ITT, intention to treat; MHM, menstrual hygiene management; NG, *Neisseria gonorrhoeae*; OR, odds ratio; RR, relative risk; SES, socioeconomic status; SSA, sub-Saharan Africa; STI, sexually transmitted infection; TV, *Trichomonas vaginalis*; VALENCIA, *VA*gina*L* community state typ*E N*earest *C*entro*I*d classifier; VMB, vaginal microbiome; WASH, water, sanitation, and hygiene.

conducted between May 2, 2018, and February 7, 2021. A total of 436 participants were included: 213 cup, 223 control. There were 289 BV diagnoses: 162 among control participants and 127 among intervention participants (odds ratio 0.76 [95% CI 0.59 to 0.98]; $p$ = 0.038). The occurrence of *Lactobacillus crispatus*–dominated VMB was higher among cup group participants (odds ratio 1.37 [95% CI 1.06 to 1.75]), as was the mean relative abundance of *L. crispatus* (3.95% [95% CI 1.92 to 5.99]). There was no effect of intervention on STIs (relative risk 0.82 [95% CI 0.50 to 1.35]). The primary limitations of this study were insufficient power for subgroup analyses, and generalizability of findings to nonschool and other global settings.

## Conclusions

Menstrual cups with BV and STI testing and treatment benefitted adolescent schoolgirls through lower occurrence of BV and higher *L. crispatus* compared with only BV and STI testing and treatment during the 30 months of a cluster randomized menstrual cup intervention.

## Trial registration

ClinicalTrials.gov NCT03051789.

---

## Author summary

### Why was this study done?

- Many girls in low- and middle-income countries are unable to adequately manage their menses and can suffer reproductive tract infections resulting from use of inappropriate materials.

- Reusable menstrual cups are medical grade silicone bell-shaped chambers that are inserted into the vagina to capture menstrual blood. Menstrual cups are safe and have not been associated with changes in vaginal pH or microflora.

- It is not known whether menstrual cups could lead to improvements in reproductive tract health.

### What did the researchers do and find?

- We assessed the impact of menstrual cups on the vaginal microbiome (VMB), bacterial vaginosis (BV), and sexually transmitted infections (STIs) in 436 secondary schoolgirls in western Kenya.

- During the 30-month cluster randomized controlled trial, BV and VMB composition were assessed every 6 months, and STIs (gonorrhea, chlamydia, and trichomoniasis) were assessed annually, with testing and treatment for BV and STIs for intervention and control participants regardless of symptoms.

- Among the intervention group, in crude analyses, the occurrence of BV was 24% lower than control participants, while the proportion of *Lactobacillus crispatus*–dominated community state type was 37% higher.

**What do these findings mean?**

- Other studies have found that menstrual cups are a safe and cost-effective tool for menstrual hygiene management.

- These results provide evidence they can promote an optimal VMB and reduce BV for adolescent girls.

- Further research should investigate the constitution of the VMB and incidence of BV and STIs in different age groups and populations using menstrual cups.

## Introduction

Adolescent girls and young women account for more than 60% of new HIV infections in sub-Saharan Africa (SSA), with an estimated 750 new infections occurring daily [1]. In western Kenya, HIV prevalence among 15- to 19-year-old adolescent girls is estimated at 3.8%, rising to 9.5% among young women aged 20 to 24 years old [2]. The sexually transmitted infection (STI) epidemic runs parallel to the HIV epidemic, with prevalences of chlamydia and gonorrhea among adolescents and young adults ranging from 7% to 17% [3–4]. Bacterial vaginosis (BV) affects 20% to 50% of the general population of women in SSA and Kenya [5] and increases the risk of HIV acquisition 1.6-fold, accounting for up to 15% of HIV infections worldwide given its high prevalence [6]. For adolescent girls, the HIV/STI epidemic overlaps with broader reproductive health concerns such as high risk of teenage pregnancy, with poor maternal outcomes, and higher risk of school dropout.

Good menstrual hygiene management (MHM) includes using clean materials to absorb or collect menstrual blood, along with adequate education, sanitation facilities, and a conducive environment [7]. Lack of MHM materials can cause adolescent girls to miss school [7] and may place them at risk for coercive sex [8–9]. To address this intersection of lack of menstrual materials and increased risk of sexual exposures, a cluster randomized study of 644 girls aged 14 to 16 years old compared the effect of provision of reusable menstrual cups to controls provided puberty and hygiene training. After 1 year, girls receiving menstrual cups had a 35% lower prevalence of BV ($p = 0.034$) and a 52% lower prevalence of STIs ($p = 0.039$) compared to controls [10]. During menses, the structure of the vaginal microbiome (VMB) (collection of microorganisms in the vagina) often shifts toward lower relative abundance of *Lactobacillus* species and an increase in *Gardnerella vaginalis*, with increased detection of BV [11]. A prior study demonstrated that menstrual cup use was not associated with vaginal microflora destabilization during menses [12], which may explain the observed beneficial effect of menstrual cups on BV and STIs.

Menstrual cup use may hold promise as a multipurpose tool for BV and STI risk reduction by maintaining a *Lactobacillus*-dominated VMB. Our objective was to study the effects of menstrual cup use on the VMB, BV, and STIs in a longitudinal analysis nested within a cluster randomized controlled trial of 4,400 secondary school girls aged 14 to 25 in Siaya County, Kenya. We hypothesized menstrual cups would lead to increase in *Lactobacillus crispatus* and reduced BV and STIs.

## Methods

### Ethics statement

This study was approved by the institutional review boards of the Kenya Medical Research Institutes (KEMRI, SERU #3215), Liverpool School of Tropical Medicine (LSTM, #15–005),

and University of Illinois at Chicago (UIC, #2017–1301). Written informed parental consent and written informed assent from minors was obtained for all participants.

## Study setting, design, and participants

Cups and Community Health (CaCHe, pronounced "Cash-Ay") was a prospective analysis of secondary school girls in Siaya County, Kenya, nested within the CCG trial (ClinicalTrials.gov NCT03051789; S1 CONSORT Checklist) [13].

The CCG trial was an open-label, 4-arm, school-cluster randomized controlled superiority trial. Schools were allocated into 4 arms (1,1,1,1) via block randomization: (1) provision of menstrual cups with training on safe cup use and care; (2) conditional cash transfer (CCT) based on ≥80% school attendance in previous term; (3) menstrual cup and CCT; and (4) usual practice. All girls received puberty and hygiene education. Girls in control and cup only arms received hand wash soap. Blinding was not possible due to the nature of the interventions (menstrual cups and cash transfer). For the CaCHe study, nested within the CCG trial, we aimed to enroll 20% of girls in the cup only and control arms of the CCG trial. Because CCG was randomized, with participants in 2 of the 4 arms included in CaCHe, it would not have been feasible, ethical, or statistically necessary to again randomize rather than maintain the randomized groups. The cup and control participants under CaCHe differed from CCG in that they all received BV and STI testing and treatment; thus, the comparison was cups plus BV and STI testing and treatment versus control plus STI testing and treatment. All cups and control schools for CaCHe were enrolled in one of the 5 wards of Rarieda subcounty. There were 16 schools screened for inclusion in Rarieda, 4 of which were excluded due to small size. Of the 12 schools, 3 schools were randomized into each arm of the CCG trial, leading to the 3 menstrual cup arm schools and 3 control arms included in CaCHe. Eligibility for CaCHe mirrored eligibility for CCG: attendance at a participating school, being a resident of the study area, provision of assent and parental/guardian consent, and girls had to report established menses (≥3 times). Girls were excluded at baseline if they declared pregnancy (*n* = 1), had not reached menarche (*n* = 4), or were under age 14 (*n* = 2) (Fig 1. Participant flow diagram). Four participants (*n* = 2 in the control arm; *n* = 2 in the menstrual cup arm) did not provide specimens for BV, VMB, or STI testing subsequent to baseline and are not included in outcome analyses. After baseline, follow-up visits occurred every 6 months through 30 months, with the 24-month visit (to have taken place April through June 2020) not conducted due to the Coronavirus Disease 2019 (COVID-19) pandemic. The cancelled 24-month visit was agreed upon by the Principal Investigator and all Co-Investigators, and aligned with IRB and ethical review requirements. Missed visits are identified in Fig 1 under "loss-to-follow-up"; the number of participants with missed visit at any time point ranged from 8 to 22 in the intervention arm and from 4 to 20 in the control arm. All participants with "missed visit" were included in analyses (only those *n* = 4 with no measurement of outcome over follow-up were excluded, as described above).

## Data collection

Participants self-completed a tablet-based survey in their language of choice (English or Dho-Luo) to obtain sociodemographic information and to assess MHM and sexual practices. Study nurses and counsellors trained in research and survey administration provided assistance as needed. Sociodemographic data included age and detailed socioeconomic measures. Socioeconomic status (SES) was measured using questions from the health and demographic surveillance system (HDSS) household survey. The SES score was constructed using the absolute index method [14], and quintiles were dichotomized as quintiles 1 to 2 and quintiles 3 to 5.

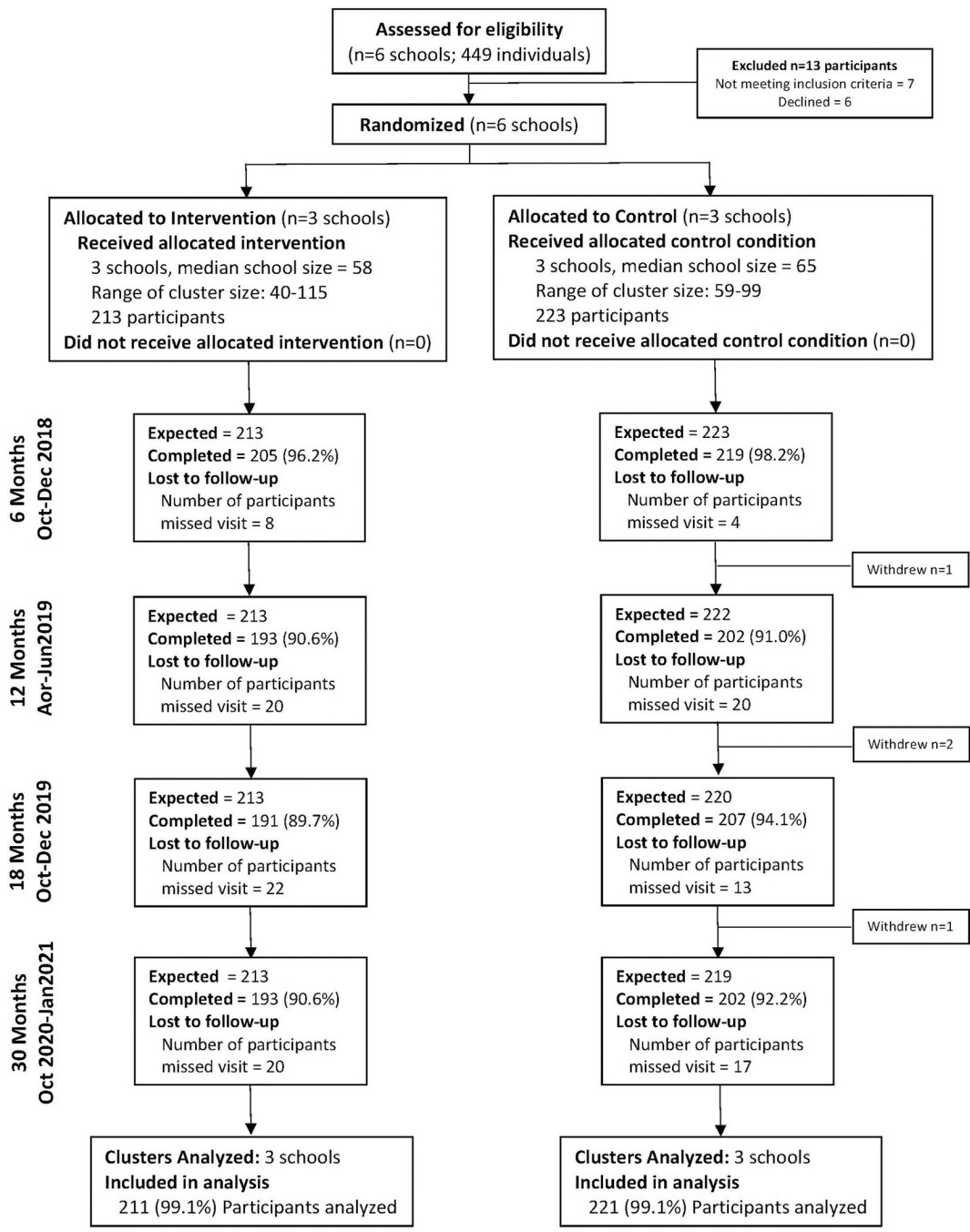

**Fig 1. Participant flow diagram for the progress of clusters and individuals through the CaCHe substudy of the Cups for Cash for Girls (CCG) cluster randomized controlled trial.**

School-level WASH (water, sanitation, and hygiene) was assessed as a score ranging 0 to 3, reflecting 1 point each for direct observation of the following: any water for handwashing, soap, and acceptable girl-to-latrine ratio (0 to 30 girls per latrine) for latrines that were clean, without offensive smell, without holes in walls, and with door, roof, and stable floor [15]. A series of questions assessed whether girls were sexually active, and whether they were forced or threatened to have sex (referred to as coerced sex), or whether they were given money, items,

or favors in exchange for sex (referred to as transactional sex). Although current pregnancy was an exclusion criterion in eligibility assessment, 3 girls who did not declare pregnancy at eligibility screening reported being pregnant after enrolment and were maintained in analyses (1 in control arm, 2 in cups arm).

## Sample size

CaCHe was designed to estimate the effect of menstrual cups on girls' risk of BV as the primary outcome, with an anticipated cumulative event rate of 30% to 40% among controls occurring over 30 months. In a design of 6 repeated measurements having AR (1) covariance structure, correlation between observations on the same subject ranging 0.25 to 0.4, and accounting for 20% loss to follow-up, group sample sizes of 220 in cup arm and 220 in control arm would achieve >80% power to detect 25% reduced relative prevalence of BV for the cup arm compared to control arm when BV prevalence is 30%, and 97% power when prevalence is 40% ($p$ = 0.05 two-sided test, 2 proportions in a repeated measures design; PASS v15).

## Randomization and masking

As detailed previously, randomization assignments were computer generated by the trial statisticians, using SAS (version 9.3), with blinded arm allocation achieved through community randomization ceremonies [13]. Participants were informed of group allocation on screening day, after which time masking the assignment was not possible. Due to the nature of the intervention, research assistants collecting specimens were not masked. Lab staff in Kenya and the United States were masked as specimens were received with only study numbers.

## Specimen collection

At baseline and each follow-up visit, girls were asked to take self-collected vaginal swabs. The first swab obtained was for 16S rRNA gene amplicon sequencing (microbiome) using the OMNIgene vaginal (OMR-130, DNA Genotek, Ontario, Canada). A cotton tipped swab was collected to test for BV. The third swab was collected for detection of *Chlamydia trachomatis* (CT) and *Neisseria gonorrhoeae* (NG) using the Cepheid GeneXpert Vaginal/Endocervical Specimen Collection Kit. The fourth swab collected was used for detection of *T. vaginalis* (TV) using rayon tipped swabs. Participants were instructed verbally and with visual aids on how to collect the swabs, as detailed elsewhere [16]. Nurses and research assistants prepared smears for BV immediately after collection. Swabs for amplicon sequencing, CT/NG, and TV were immediately placed on ice packs in coolers for transport to the lab in Kisumu. Upon receipt at the lab, specimens for amplicon sequencing were placed at −80°C until shipment to Chicago for processing.

## Detection and treatment of bacterial vaginosis and sexually transmitted infections

At baseline, swabs for CT/NG were shipped weekly for processing at the University of Nairobi Institute for Tropical and Infectious Diseases; from the 12-month visit, onward swabs for CT/NG were processed in Kisumu at Nyanza Reproductive Health Society. All swabs for TV and BV were processed in Kisumu at Nyanza Reproductive Health Society for the duration of the study. Following manufacturer protocol, vaginal swabs were tested for CT/NG using the GeneXpert (Cepheid, Sunnydale, California, United States). Swabs for TV were processed immediately upon receipt using the OSOM TV antigen detection assay (Sekisui, Lexington, MA, US). Air-dried smears were Gram stained and evaluated according to Nugent's criteria within 48

hours of receipt; a score of 7 to 10 was defined as BV [17]. For quality control, each slide was confirmed by a second technician, with discrepancies adjudicated via discussion between the 2 technicians prior to reporting. Fingerstick whole blood collected in EDTA tubes were tested for HIV according to Kenyan national guidelines. HIV–positive girls were linked to care. CT, NG, and TV were treated following Kenyan national guidelines. BV was treated with 2 g of tinidazole once daily for 2 days, for lower occurrence of gastrointestinal symptoms, likely higher adherence, and equivalent efficacy to longer duration regimens or other topical or oral antimicrobials [18–19]. All girls with Nugent score 7 to 10 were treated due to lack of correlation of reported symptoms with BV [16].

## Characterization of vaginal microbiome

DNA extraction and PCR amplification of microbial 16S rRNA genes employed a two-stage PCR protocol using primers 341F and 806R (V3-V4 variable region) [20]. Amplicons were sequenced on an Illumina MiSeq instrument, implementing V3 chemistry (600 cycles). DNA extraction, library preparation, and sequencing were performed by the Genome Research Core (GRC) at the University of Illinois Chicago. Forward and reverse reads were merged using the software package PEAR [21]. Quality filtered and primer trimmed sequence data were then processed using a standard bioinformatics pipeline for chimera removal, annotation, and community state types (CSTs) typing was conducted by University of Maryland Institute for Genomic Science [22]. Subsequently, a biological observation matrix (BIOM) was generated at the lowest taxonomic level identifiable. Vaginal CSTs were identified in a reference dataset using nearest centroid classification (*VAginaL community state typE Nearest CentroId clAssifier* (VALENCIA)) [23]. Putative contaminants were identified and removed following application of *decontam* program in R (version 4.1.3) [24]. There were 23 observations with <5,000 sequence reads, which were excluded from analyses of VMB. Prior to relative abundance estimation, data were filtered to retain taxa that contributed at least 0.01% of the total sequence reads, resulting in retention of 62 of 1,448 taxa. Taxa with highest mean relative abundance by CST and BV and STI status are summarized in S1 Fig. Raw sequence data (FASTQ files) were deposited in the National Center for Biotechnology Information (NCBI) Sequence Read Archive (SRA), under BioProject identifier PRJNA746243.

## Statistical analysis

The data analyst (RB) was blinded to group assignment. All statistical analyses were conducted using Stata/SE v17 software package. All analyses presented are intention to treat (ITT), according to assigned study arm. The study design and statistical analysis plan did not account for multiple hypothesis testing, and, therefore, results of analyses of secondary outcomes are reported as point estimates and 95% confidence intervals.

## Primary outcome

The primary outcome was BV, dichotomized as positive (Nugent score 7 to 10) versus negative (Nugent score 0 to 6). Participants in whom treatment for BV was not documented at baseline ($n = 1$) or subsequent visits ($n = 4$) were excluded from further analysis as it would be unknown whether subsequent infections represented new infections (S1 Table). Those with documented antibiotic treatment were maintained in analyses (i.e., could have multiple infections contributing to analysis; S2 Table). Generalized linear mixed models (GLMMs) were fitted to allow for the hierarchical structure of the study, with missing at random assumption for complete case data. The GLMM with binomial distribution and logit link function included treatment arm as fixed effect, and participant and cluster as random effects, to estimate odds

ratios and 95% confidence intervals, with odds ratios providing an average estimate across all time points. We estimated odds ratios for BV rather than relative risks given the relatively low efficacy of antibiotic treatment of BV, with failure in up to 50% at 6 months [25]. We applied robust variance estimate due to the small number of clusters ($n = 6$). Baseline BV and STI status were part of the longitudinal data, and, therefore, estimations account for baseline BV and STI status in both the crude and multivariable adjusted models.

Secondary outcomes were as follows: (1) VMB CST, dichotomized as CST-I (*L. crispatus* dominated) versus other CST (CST-II, CST-III, CST-IV, CST-V); (2) relative abundance of *L. crispatus* (continuous); and (3) incident STI, dichotomized as positive (composite of infection with CT, NG, and/or TV) versus negative for all three. For analysis of CST, while CST-II (*L. jesenii* dominated) and CST-V (*Lactobacillus gasseri* dominated) are generally not associated with adverse outcomes [26], they comprised a small proportion of observations and as defined a priori, the desired outcome was CST-I. Methods for analyzing CST-I followed those as described for BV. The GLMM with Gaussian distribution was applied for analysis of relative abundance of *L. crispatus*, including treatment arm as fixed effect, and participant and cluster as random effects, with 95% confidence intervals estimated via robust variance estimate. We estimated relative risk of STI given the high efficacy of antibiotic treatment for CT and NG [27–28]. The GLMM with Poisson distribution was applied for analysis of STI (as log binomial did not converge), following methods specified as above. Participants with documented antibiotic treatment for BV or STI detected at baseline or follow-up visits could contribute multiple infections (S1 and S2 Tables); one participant for whom baseline STI treatment was not documented was excluded from incidence analyses.

### Adjusted analyses and subgroup analyses

Adjusted ITT analyses were performed on the primary and secondary outcomes. Baseline a priori confounders were specified as follows: school WASH score, HIV status, age, ever sexually active, and SES. At baseline, few participants were HIV infected ($n = 7$), and due to sparsity, we did not adjust for this. ITT subgroup analyses were performed for the primary outcome (BV). The a priori specified subgroup variables were baseline values of the following: age, ever sexually active, SES, WASH score, coerced sex, transactional sex, and STI status. Subgroup analyses are presented visually as coefficient plots with 95% confidence intervals (95% CI).

### Sensitivity analyses

The 24-month endline visit was scheduled to take place April through June 2020 but did not occur due to the COVID-19 pandemic. Accordingly, the statistical analysis plan was updated so that the primary analysis included the 30-month visit, and sensitivity analysis was limited to 18 months of follow-up (i.e., prior to the COVID-19 pandemic). STI testing was to take place at annual visits (baseline, 12 months, and 24 months) but took place at 30 months due to the COVID-19 pandemic, and analyses are conducted through 30 months (i.e., there is no sensitivity analysis of STI outcome).

### Results

We randomized 3 schools to the menstrual cups only arm, and 3 schools to the control arm (Fig 1. Participant flow diagram). Recruitment took place March through April 2018, with enrollment May through June 2018. Among 442 eligible girls, 6 declined: 1 parent declined consent, and 5 minors declined assent after parent consented. Baseline characteristics were similar by study arm in relation to age, SES score, and sexual activity (Table 1; $N = 223$ control

**Table 1. Distribution of participant characteristics at baseline by study arm.**

| | Control Arm | Menstrual Cups Arm |
|---|---|---|
| | N = 223 | N = 213 |
| | n (%) | n (%) |
| Median age in years (IQR) | 17.1 (16.3–17.9) | 16.7 (15.8–17.9) |
| Age in years | | |
| 14–15 | 40 (17.9) | 62 (29.1) |
| 16 | 66 (29.6) | 60 (28.2) |
| 17 | 68 (30.5) | 44 (20.7) |
| 18 | 35 (15.7) | 31 (14.6) |
| 19–22 | 14 (6.3) | 16 (7.5) |
| Year of Schooling (Form) at enrollment | | |
| Form 1 | 0 (0.0) | 19 (9.0) |
| Form 2 | 118 (53.4) | 106 (50.0) |
| Form 3 | 103 (46.6) | 87 (41.0) |
| Missing | 2 | 1 |
| Median SES score (IQR) | 2.38 (2.14–2.63) | 2.43 (2.18–2.75) |
| Socioeconomic score, dichotomized | | |
| Quintiles 3–5 | 151 (67.7) | 158 (74.2) |
| Quintiles 1–2 | 72 (32.3) | 55 (25.8) |
| School WASH score | | |
| Score of 0 | 59 (26.5) | 115 (54.0) |
| Score of 1 or 2 | 164 (73.5) | 98 (46.0) |
| Menstrual management method[1,2] | | |
| Sanitary pads | 154 (71.6) | 155 (76.3) |
| Cloth | 7 (3.3) | 5 (2.5) |
| Sanitary pads and cloth | 54 (25.1) | 42 (20.8) |
| Missing | 8 | 11 |
| Ever sexually active: willing or coerced | | |
| No | 151 (68.6) | 136 (64.4) |
| Yes | 69 (31.4) | 75 (35.6) |
| Missing | 3 | 2 |
| Coerced sex: Ever threatened, forced, or unwanted sexual activity | | |
| No | 174 (79.1) | 154 (73.0) |
| Yes | 46 (20.9) | 57 (27.0) |
| Missing | 3 | 2 |
| Transactional sex: Ever had sex in exchange for money, items, or favors | | |
| No | 197 (90.0) | 182 (86.3) |
| Yes | 22 (10.0) | 29 (13.7) |
| Missing | 4 | 2 |
| Ever use condoms, among those ever sexually active | | |
| No | 6 (9.7) | 14 (20.6) |
| Yes | 56 (90.3) | 54 (79.4) |
| Missing | 7 | 7 |
| Number of sex partners in the past 6 months, among those sexually active | | |
| 0 | 9 (15.5) | 14 (23.3) |
| 1 | 41 (70.7) | 36 (60.0) |
| 2 | 3 (5.2) | 6 (10.0) |
| 3 to 5 | 5 (8.6) | 4 (6.7) |
| Missing | 11 | 15 |
| Number of sex partners in lifetime, among those sexually active | | |
| 1 | 45 (81.8) | 43 (68.3) |
| 2 | 5 (9.1) | 11 (17.5) |
| 3 to 8 | 5 (9.1) | 9 (14.3) |
| Missing | 14 | 12 |
| Ever been pregnant, among those sexually active | | |
| No | 56 (88.9) | 60 (88.2) |
| Yes | 7 (11.1) | 8 (11.8) |
| Missing | 6 | 7 |

(*Continued*)

**Table 1.** (Continued)

| | Control Arm | Menstrual Cups Arm |
|---|---|---|
| | *N* = 223 | *N* = 213 |
| | **n (%)** | **n (%)** |
| Currently pregnant, among those sexually active | | |
| No | 59 (98.3) | 66 (97.1) |
| Yes | 1 (1.7) | 2 (2.9) |
| Missing | 9 | 7 |
| HIV status | | |
| Negative | 216 (97.3) | 211 (99.5) |
| Positive | 6 (2.7) | 1 (0.5) |
| Missing | 1 | 1 |
| Nugent Score | | |
| Normal (0–3) | 181 (81.2) | 172 (80.8) |
| Intermediate (4–6) | 12 (5.4) | 22 (10.3) |
| BV (7–10) | 30 (13.5) | 19 (8.9) |
| BV (Nugent Score 7–10) | 30 (13.5) | 19 (8.9) |
| Any STI: Composite of *C. trachomatis*, *N. gonorrhoeae*, *T. vaginalis* | | |
| *C. trachomatis* | 24 (10.8) | 19 (8.9) |
| *N. gonorrhoeae* | 15 (6.7) | 12 (5.6) |
| *T. vaginalis* | 3 (1.4) | 3 (1.4) |
| | 9 (4.0) | 4 (1.9) |
| CST | | |
| CST-I (*L. crispatus* dominated) | 93 (42.9) | 93 (44.1) |
| CST-II (*L. jensenii* dominated) | 5 (2.3) | 7 (3.3) |
| CST-III (*L. iners* dominated) | 73 (33.6) | 72 (34.1) |
| CST-IV (mixed) | 43 (19.8) | 34 (16.1) |
| CST-V (*L. gasseri* dominated) | 3 (1.4) | 5 (2.4) |
| Missing | 6 | 2 |
| Distribution of relative abundance of *Lactobacillus crispatus* | | |
| 0% (absent) | 55 (25.4) | 49 (23.2) |
| >0%–<25% | 65 (30.0) | 61 (28.9) |
| 25%–<50% | 20 (9.2) | 24 (11.4) |
| 50%–<75% | 13 (6.0) | 22 (10.4) |
| 75%–100% | 64 (29.5) | 55 (26.1) |
| Missing | 6 | 2 |
| Mean percent relative abundance of *L. crispatus* (SD) | 34.4 (39.4) | 35.4 (35.4) |

BV, bacterial vaginosis; CST, community state type; IQR, interquartile range; SD, standard deviation; SES, socioeconomic status; STI, sexually transmitted infection; WASH, water, sanitation, and hygiene.

[1]There were 11 participants who reported tampon use (*n* = 5 intervention arm; *n* = 6 control arm), who were also using pads (*n* = 4), cloth (*n* = 1), and pads and cloth (*n* = 6).

[2]There were 5 participants who reported menstrual cup use (*n* = 2 intervention arm; *n* = 3 control arm), who were also using pads (*n* = 1) and pads and cloth (*n* = 4).

arm, *N* = 213 menstrual cup arm), but with large imbalance in WASH scores, being higher among participants in the control schools. Baseline STI and CST were similar between groups, and BV was somewhat increased among control arm participants. Participants were median 16.9 years of age (IQR 16.1 to 17.9), with 11.2% having BV and 9.9% with any STI. Follow-up visits occurred as planned with follow-up of >90% of enrolled participants at each study visit.

Overall, the prevalence of BV and STI increased over time, and the proportion of participants with CST-I and the mean relative abundance of *L. crispatus* decreased over time (Table 2 and Fig 2). The prevalence of BV was greater among control arm participants than cup arm participants at all follow-up time points except at 18 months. The prevalence of STI was similar between arms at 12 months, and with higher rates in control arm participants compared to

**Table 2. BV, VMB, and STI outcomes by treatment arm and study visit.**

| | Control Arm | Menstrual Cup Arm | Total |
|---|---|---|---|
| | n/N (%) | n/N (%) | n/N (%) |
| BV | | | |
| 6 months | 21/219 (9.59) | 18/205 (8.78) | 39/424 (9.20) |
| 12 months | 35/202 (17.3) | 22/193 (11.4) | 57/395 (14.4) |
| 18 months | 27/207 (13.0) | 29/191 (15.2) | 56/398 (14.1) |
| 30 months | 49/202 (24.3) | 39/193 (20.2) | 88/395 (22.3) |
| STI | | | |
| 12 months | 23/202 (11.4) | 23/193 (11.9) | 46/395 (11.7) |
| 30 months | 38/202 (18.1) | 26/193 (13.5) | 64/395 (16.2) |
| CST-I vs. other CST | | | |
| 6 months | 90/219 (41.1) | 89/202 (44.1) | 179/421 (42.5) |
| 12 months | 66/195 (33.9) | 80/188 (42.6) | 146/383 (38.1) |
| 18 months | 73/204 (35.8) | 81/190 (42.6) | 154/394 (39.1) |
| 30 months | 58/202 (28.7) | 67/192 (34.9) | 125/394 (31.7) |
| Mean relative abundance of *Lactobacillus crispatus* | **Percent (SD)** | **Percent (SD)** | **Percent (SD)** |
| 6 months | 31.1 (31.1) | 32.4 (36.1) | 31.8 (35.9) |
| 12 months | 25.2 (33.4) | 29.7 (32.9) | 27.4 (33.2) |
| 18 months | 26.3 (34.6) | 34.2 (35.9) | 30.1 (35.4) |
| 30 months | 22.6 (34.6) | 27.8 (36.2) | 25.1 (35.4) |

BV, bacterial vaginosis; CST, community state type; SD, standard deviation; STI, sexually transmitted infection; VMB, vaginal microbiome.

menstrual arm participants (18.1% versus 13.5%) at 30 months (Table 2). The proportion of participants with CST-I and the mean relative abundance of *L. crispatus* were higher for cup arm participants than control arm participants at all follow-up points.

There was evidence of a beneficial intervention effect on the primary outcome, BV. Compared to the control arm, the odds of BV in the menstrual cup arm was 26% lower (odds ratio [OR] 0.74 [95% CI 0.59 to 0.98]; $p = 0.038$; Table 3). There were also beneficial effects on secondary outcomes, with 37% increased odds of CST-I for participants in the cup arm (OR 1.37 [95% CI 1.06 to 1.75]) and relative abundance of *L. crispatus* (mean absolute difference 3.95% [95% CI 1.92 to 5.99%]). There was no evidence of intervention effects on STIs (relative risk (RR) 0.82; 95% CI: 0.50 to 1.35).

## Adjusted, subgroup, and sensitivity analyses

Controlling for a priori specified confounders (age, sexually active at baseline, SES, and school WASH score), the associations between menstrual cup arm and BV, CST-I, and *L. crispatus* were largely similar, though the association between cup arm and BV was no longer statistically significant (Table 3). In adjusted analyses, there was significant benefit against STIs in the cup arm (adjusted RR 0.77 [95% CI, 0.62 to 0.95]). In subgroup analyses, the protective effect of menstrual cups against BV showed modest variation by age, SES score, being sexually active at baseline, ever being coerced into sex, ever engaging in transactional sex, and STI status (Fig 3). However, confidence intervals were wide due to reduced samples sizes within strata, and for most subgroup estimates, 95% CIs included 1. In sensitivity analyses (Table 4), primary outcome and secondary outcomes were examined through 18 months (i.e., prior to the COVID-19 pandemic). The crude point estimate for the effect of menstrual cups on BV through 18 months was the same as through 30 months (OR 0.74). The beneficial effect of

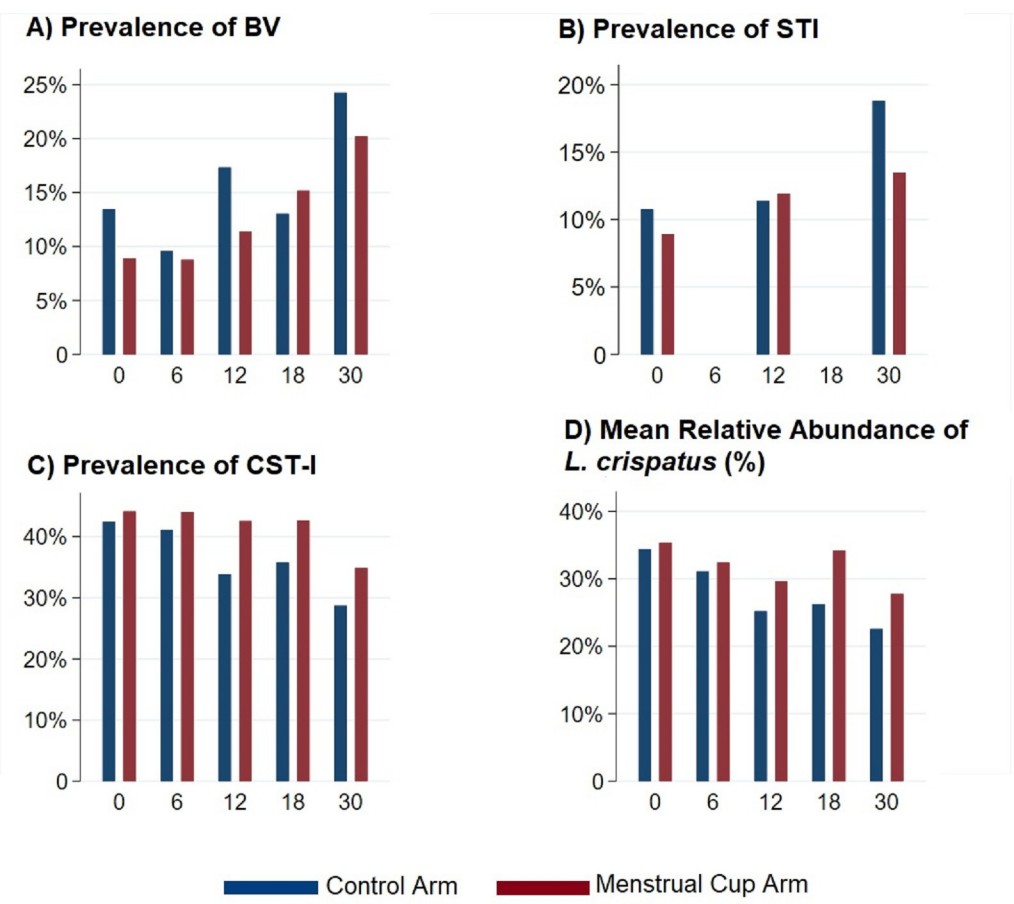

**Fig 2. Bar chart of prevalence of BV, STI, CST-I, and mean relative abundance of *L. crispatus* by randomization status over time**. The figure shows the prevalence of (**A**) BV, (**B**) STI (composite of *C. trachomatis*, *N. gonorrhoeae*, and *T. vaginalis*), (**C**) CST-I (*L. crispatus* dominated), and (**D**) mean relative abundance of *L. crispatus*, by intervention status over study visit in months. Measures from participants in the control arm are depicted in navy bars, and from participants in the menstrual cup arm in maroon bars. Study visit in months is depicted on the x-axis. BV, bacterial vaginosis; CST-I, community state type I; STI, sexually transmitted infection.

menstrual cups on CST-I versus other CST was similar (OR 1.35 through 18 months versus OR 1.37 through 30 months). The effect of cups of relative abundance of *L. crispatus* also remained similar when excluding the 30-month visit.

There was one adverse event in which a participant was unable to remove her cup (cup retention), which occurred between the 6- and 12-month study visits and required nurse assistance for removal.

## Discussion

In this prospective analysis nested within a cluster randomized controlled trial, we observed via ITT analysis that secondary schoolgirls allocated to schools receiving the menstrual cup with BV and STI testing and treatment had a 26% lower odds of BV, 37% increased odds of having optimal VMB CST (CST-I), and 3.95% increased relative abundance of *L. crispatus*, compared with controls without cups, who also had BV and STI testing and treatment. Effect sizes did not vary when adjusting for a priori specified confounders. We observed a statistically significant lower incidence of STI for participants in the menstrual cup arm only when

**Table 3. Results of crude and multivariable adjusted analyses: Effect of menstrual cups on BV, STI, CST-I, and relative abundance of *Lactobacillus crispatus*.**

|  | Crude | Multivariable Adjusted* |
|---|---|---|
| **BV** | **OR (95% CI)** | **OR (95% CI)** |
|  | N = 2,048 | N = 2,025 |
|  | 0.76 (0.59–0.98), p = 0.038 | 0.82 (0.51–1.32), p = 0.421 |
| **STI** | **RR Ratio (95% CI)** | **RR Ratio (95% CI)** |
|  | N = 1,226 | N = 1,212 |
|  | 0.82 (0.50–1.35) | 0.77 (0.62–0.95) |
| **CST-I vs. other CST** | **OR (95% CI)** | **OR (95% CI)** |
|  | N = 2,024 | N = 2,001 |
|  | 1.37 (1.06–1.75) | 1.42 (1.21–1.67) |
| ***Lactobacillus crispatus*** | **Mean Relative Abundance (95% CI)** | **Mean Relative Abundance (95% CI)** |
|  | N = 2,024 | N = 2,001 |
|  | 3.95 (1.92–5.99) | 4.46 (2.76–6.16) |

BV, bacterial vaginosis; CST, community state type; CST-I, community state type I; OR, odds ratio; RR, relative risk; SES, socioeconomic status; STI, sexually transmitted infection; WASH, water, sanitation, and hygiene; 95% CI, 95% confidence interval.

*Adjusted for: baseline age in years, baseline SES score, baseline WASH score, baseline reported sexual activity status, and time.

adjusted for confounders. Sensitivity analyses restricted to pre-COVID data through 18 months of follow-up observed relative increases in CST-I and *L. crispatus* that were similar to associations demonstrated through 30 months of follow-up. This suggests that the effect of menstrual cups on VMB was not altered during the COVID-19 pandemic time period (i.e., the 30-month visit), despite overall increasing prevalence of BV and STIs.

Destabilization of the vaginal microbiome composition during menses is well documented, with increases in *G. vaginalis* and decreases in *L. crispatus* [11,29–31]. There is biologic plausibility for menstrual cups supporting an optimal VMB composition and preventing destabilization. Increased iron levels during menses are favorable to *G. vaginalis* [32], a major pathobiant in BV [17,33–34]. Because blood is collected inside the menstrual cup, there is limited blood flow in the vaginal vault. Additionally, the vaginal pH increases during menses [11], due to decreases in acidifying *Lactobacilli* during menses and also because blood pH is typically around 7. Among 406 women observed over 3 menstrual cycles of menstrual cup use, North and colleagues observed no changes in vaginal pH, appearance of vulva or cervix (e.g., ectopy, friable, other abnormal), or detection of *Candida*, yeast cells, or *T. vaginalis* [12]. In a subset of 44 women in this study, colposcopy conducted at baseline, 2 to 3 months, and 5 to 6 months yielded no change in inflammation, abrasion, punctuation, acetowhite findings, or topography. In these same women, there was no change in colonization detected via culture for *Lactobacillus* spp., *G. vaginalis*, *Bacteroides* spp., or *E. coli*. Our findings build on these studies, showing that menstrual cup use can benefit the VMB and reduce BV, measured using sensitive molecular methods, in the setting of a rigorous cluster randomized controlled study design.

Poor-quality MHM products have been associated with reproductive tract infections. The mechanisms are biological and behavioral. Unsanitary cloths, disposable pads used beyond their recommended time, or reusable pads that are not sufficiently washed and dried are associated with increased reports of vaginal discharge, reproductive tract infections, and urinary tract infection symptoms [35]; this may occur from direct transfer of bacteria or the occlusive environment. Additionally, adolescent girls and young women with insufficient access to

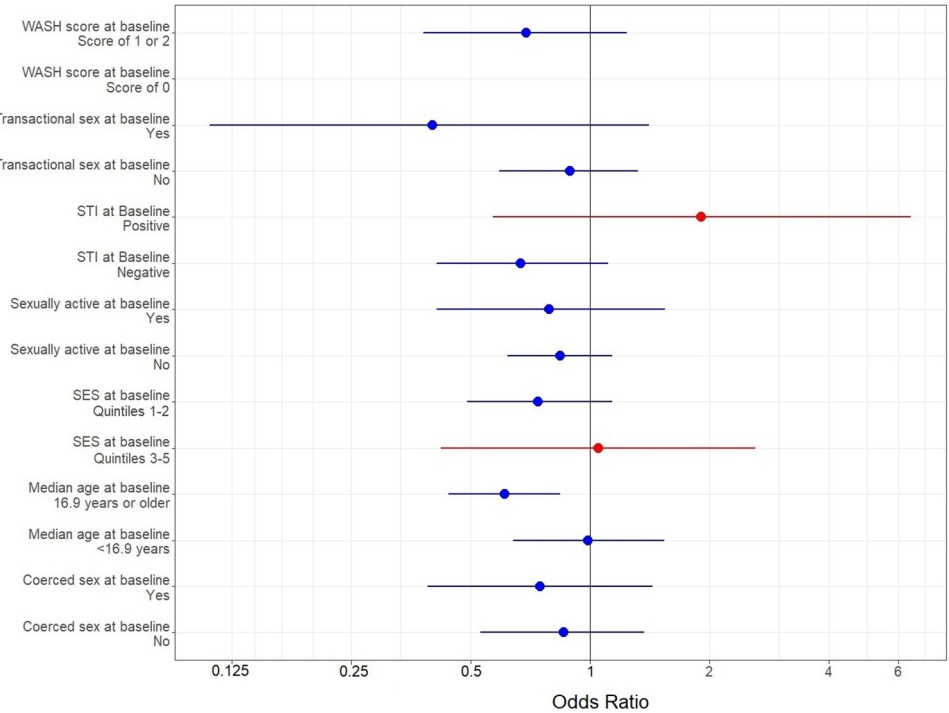

**Fig 3. Coefficient plot summarizing the results of subgroup analysis of primary outcome, BV.** The coefficient plot shows the association between menstrual cups arm and BV, among strata of prespecified subgroups: median age, SES, WASH, sexually active at baseline, coerced sex at baseline, transactional sex at baseline, and baseline STI (composite of *C. trachomatis*, *N. gonorrhoeae*, and *T. vaginalis*). The x-axis shows the value of the coefficient with subgroups listed on the y-axis. The point estimate is represented by the central circle. The bars extending from the point estimate represent the 95% CI. Positive associations (OR greater than 1) are shaded red, and inverse associations (OR less than 1) are shaded blue. The reference y-line is indicated at 1. For participants with WASH score of 0, this was collinear with cluster and could not be estimated. BV, bacterial vaginosis; OR, odds ratio; SES, socioeconomic status; STI, sexually transmitted infection; WASH, water, sanitation, and hygiene; 95% CI, 95% confidence interval.

MHM products may be vulnerable to transactional sex to obtain these materials. In Phillips-Howard's household study of 3,418 menstruating girls and women aged 13 to 29 years in the same area of western Kenya as our present study, 10% of 15-year-olds reported they exchanged sex for sanitary pads [8]. We observed that crude point estimates for BV, CST-I, and *L. crispatus* associated with the menstrual cups arm were unchanged when adjusted for baseline age, SES, WASH score, and sexual activity, suggesting the mechanism by which menstrual cups conferred benefit in our sample was independent of these sociobehavioral factors.

**Table 4. Results of sensitivity analysis: Excluding 30-month visit for BV, CST-I vs.** other CSTs, and relative abundance of *Lactobacillus crispatus*.

|  | Crude OR (95% CI) |
|---|---|
| **BV** (*N* = 1,653) | 0.74 (0.41–1.32) |
| **CST-I vs. other CST** (*N* = 1,630) | 1.35 (1.10–1.66) |
|  | **Mean Relative Abundance (95% CI)** |
| ***Lactobacillus crispatus*** (*N* = 1,630) | 3.51 (1.17–5.75) |

BV, bacterial vaginosis; CST, community state type; CST-I, community state type I; OR, odds ratio; 95% CI, 95% confidence interval.

The COVID-19 pandemic caused the 24-month endline visit to be delayed, thus introducing potential confounding and bias. We observed substantial increases in BV and STI at the 30-month visit (i.e., including time during the COVID-19 pandemic), and we also observed meaningful and statistically significant benefits conferred to VMB composition and against BV occurrence through 18 months of follow-up time, occurring completely prior to the COVID-19 pandemic. The effect sizes were similar between data including the 30-month visit and data excluding the 30-month visit, suggesting lack of confounding. After trial end, all participants have received menstrual cups and follow-up is ongoing to 72 months. This extended follow-up will enable us to observe whether the changes associated with menstrual cups vary by changes in sexual activity (such as number of sexual partners, condom use, and coerced and transactional sex) and a greater cumulative incidence and prevalence of past STIs and BV.

## Strengths and limitations

Nesting our study of menstrual cups and the effect on BV, the VMB, and STIs within the CCG cluster randomized controlled trial with ITT analyses minimized confounding and selection bias and maximized efficient use of resources. We did not directly measure the hypothesized mechanisms by which menstrual cups may benefit the VMB; specifically, we did not intensely measure the VMB composition, bacterial load, or vaginal pH across the menstrual phases by MHM product. While infeasible in our setting, future studies should examine these mechanisms as they may lead to development of other interventions that benefit the VMB. Our future analyses will evaluate the microbiome of menstrual cup specimens obtained in this study, to determine correlation to host microbiome and detection of putative pathogens. Condom use and number of sex partners may have differed between groups, but as these questions were answered only by the subset of participants reporting sexual activity, we could not adjust for these potential confounders. It is possible certain users may benefit differentially from menstrual cup use, but the study was not powered for such subgroup analyses, and we refrain from interpretation; potential differences must be assessed in future scaled assessments of menstrual cup impact on BV and STIs. We believe our findings generalize to adolescent girls attending secondary schools, and results for those who are not in school or in different global regions are unknown. We note both in the cup arm and in the control arm, participants underwent BV and STI testing and treatment at scheduled study visits. Though both arms were tested and treated equivalently for BV and STIs, we treated BV (Nugent score 7 to 10) regardless of symptoms due to lack of correlation with diagnosis [16], and this is not standard of care for BV, and, therefore, results may not generalize to those not screened or treated for BV.

## Conclusions

These results generated from a randomized study design suggest beneficial effects of menstrual cups on the composition of the VMB and reduction in BV. This evidence should be used in the development and implementation of MHM programs. In tandem with cost-effectiveness and reduced environmental impacts [35–36], beneficial impact on reproductive tract health supports menstrual cups as a priority menstrual product in MHM programs.

## Supporting information

**S1 CONSORT Checklist. Checklist of items that should be included in reports of randomized studies.**
(PDF)

**S1 Fig. Stacked bar chart showing relative abundance of 10 taxa with highest mean relative abundance by CST for each participant.** The relative abundance of the 10 taxa with the highest mean relative abundance is shown (y-axis), separated by CST (x-axis) with individual subjects represented by individual bars for observations from (**A**) control arm participants and (**B**) intervention arm participants. The bar at the top of each graph represents the presence of BV and/or STI for each observation. BV, bacterial vaginosis; CST, community state type; STI, sexually transmitted infection.
(TIF)

**S1 Table. Number of tests conducted, infections detected, and documented antimicrobial treatment by study time point.**
(DOCX)

**S2 Table. Cumulative number of BV and STI cases by study arm.**
(DOCX)

**S1 Statistical Analysis Plan. Statistical analysis plan.**
(PDF)

## Author Contributions

**Conceptualization:** Supriya D. Mehta, Penelope Phillips-Howard.

**Data curation:** Garazi Zulaika, Anna Maria van Eijk, Fredrick Otieno.

**Formal analysis:** Runa Bhaumik.

**Funding acquisition:** Supriya D. Mehta, Penelope Phillips-Howard.

**Investigation:** Supriya D. Mehta, Runa Bhaumik, Stefan J. Green, Penelope Phillips-Howard.

**Methodology:** Supriya D. Mehta, Garazi Zulaika, Walter Agingu, Elizabeth Nyothach, Runa Bhaumik, Stefan J. Green, Anna Maria van Eijk, Fredrick Otieno.

**Project administration:** Garazi Zulaika, Walter Agingu, Elizabeth Nyothach, Daniel Kwaro, Fredrick Otieno, Penelope Phillips-Howard.

**Resources:** Supriya D. Mehta, Walter Agingu, Elizabeth Nyothach, Stefan J. Green, Daniel Kwaro, Fredrick Otieno, Penelope Phillips-Howard.

**Supervision:** Supriya D. Mehta, Garazi Zulaika, Walter Agingu, Elizabeth Nyothach, Stefan J. Green, Daniel Kwaro, Fredrick Otieno, Penelope Phillips-Howard.

**Visualization:** Supriya D. Mehta, Runa Bhaumik.

**Writing – original draft:** Supriya D. Mehta.

**Writing – review & editing:** Garazi Zulaika, Walter Agingu, Elizabeth Nyothach, Runa Bhaumik, Stefan J. Green, Anna Maria van Eijk, Daniel Kwaro, Fredrick Otieno, Penelope Phillips-Howard.

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
