## [Editor Report · Decision Letter 0]

9 Nov 2022

Dear Dr Mehta, 

Thank you for submitting your manuscript entitled "The effect of menstrual cups on the vaginal microbiome, Bacterial vaginosis, and sexually transmitted infections: results of a nested cohort study within a cluster randomized controlled trial" for consideration by PLOS Medicine.

Your manuscript has now been evaluated by the PLOS Medicine editorial staff as well as by an academic editor with relevant expertise and I am writing to let you know that we would like to send your submission out for external peer review.

Please re-submit your manuscript within two working days, i.e. by Nov 11 2022 11:59PM.

Kind regards,

Callam Davidson

Associate Editor

PLOS Medicine

---

## [Decision Letter · Decision Letter 1]

9 Jan 2023

Dear Dr. Mehta,

Thank you very much for submitting your manuscript "The effect of menstrual cups on the vaginal microbiome, Bacterial vaginosis, and sexually transmitted infections: results of a nested cohort study within a cluster randomized controlled trial" (PMEDICINE-D-22-03601R1) for review by PLOS Medicine. Your paper was evaluated by a senior editor and discussed among all the editors here. 

Your article was also evaluated by an academic editor and three independent reviewers. I am afraid that the reviewers raised a number of substantial concerns about the paper. After discussion among the editorial team, I am sorry to say that we have decided not to consider it further for publication in the journal.

The reviews are included below or at the following link: [LINK]. I hope that you find them constructive. 

I am sorry that I cannot be more positive on this occasion. I hope you appreciate the reasons for this decision, and will consider PLOS Medicine for other submissions in the future. 

Best wishes,

Callam Davidson, 

PLOS Medicine

plosmedicine.org

Reviewer Notes:

Reviewer #1: Statistical review

This paper reports a substudy of a cluster randomised trial that examines whether menstrual cups reduce the prevalence of bacterial vaginosis. The authors demonstrate that there is a significant reduction in the primary analysis.

Generally the study used appropriate methods and was reported well. I have some comments below:

1. Abstract + results page 14: I would recommend providing the p-values for the secondary outcomes in addition to the CIs.

2. Page 8, sample size calculation: is the 25% reduced prevalence in absolute terms (i.e. from 30% to 5%) or in relative terms? Was any clustering effect allowed for?

3. Statistical methods: the follow-up rate was generally quite high, but it would be good to mention what assumptions about missing data were made (i.e. that the analysis model allowed for missing at random patterns of missingness).

4. Statistical methods: can the authors provide some intuition about how to interpret the estimated odds ratio - is this basically an average across all time points?

5. Page 14: I feel that the text should point out that the multivariable adjusted model for the primary outcome no longer was significant (especially as the authors highlight a secondary endpoint becoming significant with the adjustment).

James Wason

Reviewer #2: This manuscript presents the results of a cohort study nested within a randomized trial examining the effect of menstrual cups compared to usual care on bacterial vaginosis in a population of adolescent girls and young women. The authors conclude that menstrual cups benefitted adolescents by reducing BV and increasing lactobacillus crispatus relative abundance. There are some critical analytical problems that lead me to seriously question the primary findings. In addition, the authors discuss the findings as though they are derived from a RCT. The net effect is that they make strong statements about the benefits of the menstrual cup based on findings that do not seem justified by the data presented.

Major Comments

1) Methods state that "All girls with Nugent score 7-10 were treated due to lack of correlation of reported symptoms with BV." This rationale does not justify treating asymptomatic BV - the only indication for treatment of asymptomatic BV is in women who are pregnant and have a history of preterm birth. Even this indication is currently being questioned. Equally importantly, the decision to treat all BV in this cohort means that the comparison groups were actually menstrual cup plus regular screening and treatment of BV versus regular screening and treatment of BV. It does not really address the question of whether menstrual cup alone reduces BV in a realistic setting where only symptomatic BV is treated.

2) The results state that baseline BV, STI, and CST were similar between groups. This is not true for BV, the primary outcome of the trial. In fact, the BV prevalence in the control arm was 13.5% (30/223) while the prevalence of BV was 8.9% (19/213) in the cup arm. By a quick calculation, this appears to yield an odds ratio of 0.66 for BV in the cup versus control arm prior to the intervention - bigger than the effect seen later in the study. Since the authors describe this as a cohort study, it seems like they should adjust for this baseline difference in the primary outcome variable. Instead, the analysis was conducted as though this was an RCT. Even if you consider it to be a small cluster RCT with too few clusters, it would be important to adjust for a baseline difference if the randomization did not result in similar rates of the outcome in the two arms.

3) Discussion states, "Our findings build on these studies, showing that menstrual cup use can benefit the vaginal microbiome and reduce BV, measured using sensitive molecular methods, in the setting of a rigorous cluster randomized design." This is misleading - the analysis presented in this paper is a cohort (or a cRCT with too few clusters), and did not account for an important baseline difference in the outcome.

Reviewer #3: Mehta and colleagues used data from a cluster randomized trial conducted among secondary schools in Kenya to evaluate the effect of menstrual cup use, versus normal menstrual management practices, on BV, the vaginal microbiome, and STI. Their data indicate that individuals attending schools randomized to receive menstrual cups had 24% reduced odds of BV and 37% higher odds of L. crispatus-dominated microbiota. These novel findings warrant replication in additional populations, but they suggest provision/promotion of menstrual cups may be a relatively easy-to-implement intervention to improve vaginal microbiota composition and reduce BV. Overall, the manuscript is well written and clear. I'll share some suggestions with the goal of improving reader experience and the dissemination of these important and exciting findings.

Major comments

1. In the Methods, please provide details on Gram stain and Nugent score quality control and quality assurance measures. With BV assessed by Nugent score as the primary outcome, understanding the efforts taken to reduce inter-observer variation and improve accuracy in Nugent scoring is essential to evaluating the validity of the findings.

2. In the Methods, antibiotic-use-related inclusion and exclusion criteria are unclear and presented/described differently in several places (lines 164-165, 193-196, 211-214). The rationale for these criteria is also unclear. Please more clearly describe the criteria and why they were used (I think to allow for repeat outcomes following antibiotic-mediated BV/STI clearance?), and please include all relevant information together in one place.

3. A figure depicting participants' baseline and end-of-follow-up microbiota composition (e.g. stacked bar plot, heatmap), indicating whether participants attended schools in the menstrual cup versus control arm, and indicating presence of BV and STI would be very helpful in conveying the microbial context of the trial and its findings. Please consider including such a figure.

4. The first paragraph of the Results seems to undersell/under-report some of the differences in baseline characteristics between the intervention arms. It is somewhat subjective what constitutes a small vs. large difference, but here are some comments from the text I'd suggest reconsidering/revising: 

a. "some imbalance in…school WASH score" (line 247) implies minor differences. However, about half of menstrual cup schools had a score of 0 compared to about a quarter of control arm schools. 

b. "Baseline BV…were similar" (line 248). Similar proportions of participants had Nugent score 0-3 at baseline; however, the prevalence of Nugent score 4-6 was two-fold higher in menstrual cup schools, and the prevalence of Nugent score 7-10 was about 1.5-fold higher in control arm schools.

c. Among ever-sexually-active participants, condomless sex/never using condoms was 2-fold higher among menstrual cup schools. This may be worth mentioning in the text. 

5. I think the subgroup analysis stratified by history of transactional sex deserves more attention in the discussion (because its exciting)! The BV OR was substantially lower in magnitude among those reporting a history of transactional sex than among those reporting never engaging in transactional sex. Neither OR is significant, and the CI for those reporting transactional sex is wide, so it will be important to not over interpret these findings. That said, this difference and the data suggesting a potentially stronger protective effect among those who do report history of transactional sex directly speak to one of the hypothesized sociobehavioral mechanisms by which menstrual cup use may reduce BV and improve vaginal health - that menstrual cups eliminate (at least one) cause/motivation to engage in transactional sex, which is to be able to afford/acquire menstrual management materials.

6. Not reporting data on either baseline menstrual management practices or follow-up adherence to menstrual cup use among those attending menstrual cup schools are important limitations that should be discussed. Alternatively, if these data are available, please include them in the descriptive statistics.

Minor comments

1. The hypothesis that menstrual cup use may reduce reliance on transactional sex to afford/acquire menstrual management materials (lines 62-63) should be explicitly stated earlier in the introduction.

2. The language used to describe socioeconomic status categories (line 110) is stigmatizing, especially considering you refer to both categories as "poor." I would recommend using more a quantitative description (e.g. just referring to the cutpoint between quintiles 2 and 3). Please also be sure to use the same language to describe these categories in the Methods, Table 1, and Figure 3.

3. Please clarify whether the "water for handwashing" component of the WASH score (line 112) refers to clean water or any water.

4. Throughout the methods, please provide the version number for all software used.

5. Please describe the design of study follow-up in the text of the Methods (e.g. the number and timing of follow-up visits).

6. In lines 153-155, the order and timing of which institutes received and processed swabs for BV and STI detection is unclear. Please revise.

7. Please state the cutpoints/strata for subgroup analyses in the Methods (lines 224-226).

8. Results paragraph in lines 271-277 should refer to Table 3.

9. There are a few places that would benefit from additional references:

a. Line 324 - more than one reference should be provided as microbiota destabilization during menses is, as you state, well documented.

b. Line 327 - please provide a reference(s) for iron being favorable to Gardnerella and a primary/original research reference for Gardnerella being a major BV pathobiont (not just a review).

c. Lines 342-345 - again, please provide primary/original research reference(s) for poor quality menstrual management materials being associated with adverse outcomes.

10. It is important to note that these findings may not be generalizable to individuals attending secondary school in regions other than East Africa, sub-Saharan Africa, especially considering typical menstrual management practices are often "passed down" matrilineally and likely differ between global regions.

11. Please place the menstrual cup arm and control arm columns in the same order in Table 1, Table 2, and Supplemental Table 2.

12. In Figure 3, please include axis tick marks and labels for ORs <1. Please also clarify in the caption whether a non-linear scale has been applied to the x axis.

Signed: Kayla A. Carter

Reviewer #4: This is a well-conducted study on an important topic to improve vaginal health in adolescent girls and young women in a setting with high HIV and STI burden. The study appears to be rigorous in its rationale, design, and analysis. I have only minor revisions and suggestions regarding the analysis.

1. Is it worth carrying out another sensitivity analysis that leaves out BV-intermediate? Lumping BV intermediate in with BV negative could be diluting the effects, especially given the L. crispatus finding is statistically stronger. Perhaps BV negative could be compared to BV positive, or BV positive and intermediate together to BV negative. A lot of the BV intermediate could be L.iners dominated.

2. There is analysis of the absolute abundance of L.crispatus. It would be of interest to know what the menstrual cup is doing to overall bacterial load, in addition to relative abdunance.

3. Should relative abundance of L.crispatus vs L.iners vs non-lacto be considered?

4. The authors could specify the number and intervals of visits in the design. At the end of the methods and in the figures, this appears to be 6-monthly, but the number of observations per participant is an important aspect of the study. 

5. the difference in unadjusted vs adjusted STI effect is puzzling. Why do the 95% CI vary so widely (wide in unadjusted but narrower in adjusted, with a similar effect size?) Similarly, the 95% CI for the BV effect are narrow in unadjusted but quite wide in the adjusted, despite similar effect sizes for menstrual cups vs BV - it would be useful to know what the key variables are that are changing these outcomes so dramatically. 

6. Were any interaction analyses considered for Fig3? Even if non-significant these could be worth reporting, given some of the differences between strata.

7. There is no comment in the discussion on cost-effectiveness versus effect size- do these results favour scaling up this intervention? Would the increasing availability of reusable cups favour their widespread distribution? While its clearly beyond scope to do this analysis here, some speculation re: future directions would be useful.

[LINK]

---

## [Decision Letter · Decision Letter 2]

14 Mar 2023

Dear Dr. Mehta,

Thank you very much for submitting your revised manuscript "Lower prevalence of Bacterial vaginosis and increased relative abundance of Lactobacillus crispatus associated with menstrual cups: results of a nested cohort study within a cluster randomized controlled trial" (PMEDICINE-D-22-03601R2) for consideration at PLOS Medicine. 

Your paper was discussed again with the academic editor, and sent back to independent reviewers, including a statistical reviewer. The reviews are appended at the bottom of this email and any accompanying reviewer attachments can be seen via the link below:

[LINK]

In light of these reviews, I am afraid that we will still not be able to accept the manuscript for publication in the journal in its current form, but we would like to consider a revised version that addresses the reviewers' and editors' comments. Obviously we cannot make any decision about publication until we have seen the revised manuscript and your response, and we plan to seek re-review by one or more of the reviewers. 

We hope to receive your revised manuscript by Apr 04 2023 11:59PM. Please email us (plosmedicine@plos.org) if you have any questions or concerns.

We look forward to receiving your revised manuscript. 

Sincerely,

Callam Davidson, 

PLOS Medicine

plosmedicine.org

Comments from the Academic Editor:

1. Change in title to better reflect study design.

2. Description of the intervention as cup plus screening and treatment of BV versus screening and treatment of BV.

3. Clear statement regarding generalisability of the findings in the discussion.

Please respond to the comments from Reviewers 2 and 3. 

Did your sub-study have its own associated prospective protocol or analysis plan (separate from the trial protocol available at https://bmcpublichealth.biomedcentral.com/articles/10.1186/s12889-019-7594-3)? The trial protocol and CT registry contain little to no information regarding the present sub-study, and it is difficult to determine how much of the study was pre-specified based on the information currently available.

Please confirm whether the primary results of the trial have been published.

As your trial had to undergo important modifications in response to extenuating circumstances, please complete the CONSERVE-CONSORT checklist and provide in your Supporting Information.

Comments from the reviewers:

Reviewer #1: Thank you to the authors for addressing my previous comments well. I have no further issues to raise.

Reviewer #2: The revised paper has the same issues as the earlier submission.

1) Asymptomatic women with a Nugent score >=7 do not have a disease. There is no study showing any benefit to treating women (or adolescents) in this scenario. Regardless of what the study team thought, this is not the standard of care. I can understand that if the IRB were presented with the information that it was a disease, they would want it treated. If instead the IRB understood that there is no benefit to treatment of asymptomatic women with a vaginal gram stain score >=7, I suspect they would have felt differently. 

2) Because the research team's approach to handling asymptomatic women with a Nugent score >=7 this is clearly a deviation from the standard of care, I think this needs to be clearly stated in the paper.

3) It is true that both arms were tested and treated equally for BV, but the response that this does not introduce bias misses the point. It is a problem with generalizability. The study did not test menstrual cup versus no menstrual cup. Instead, it tested regular screening and treatment for BV (regardless of symptoms) versus regular screening and treatment of BV plus a menstrual cup. It is not possible to generalize that the effect on BV would be the same in the absence of regular screening and treatment of BV in both arms. I think that for the paper to be publishable, it would need to

a. Clearly explain the two arms as: (1) cup plus screening and treatment of BV versus (2) screening and treatment of BV

b. In the discussion, clearly note that it may or may not be possible to generalize these findings to a population that is not receiving regular screening and treatment for both asymptomatic and symptomatic BV. I.e. without doing the study, you cannot assume that menstrual cups alone would have the same effect. It might be greater or less than the effect observed in this trial.

4) I'm puzzled by the authors response about randomization, given that the title of the study presents it as a nested cohort study within a cluster-randomized trial. The title appears to present it as a cohort. I actually agree that the randomization remains valid, so it may be the title that is the problem. Of course it would not make sense to re-randomize. However, it is a very small cRCT - I would still like to see the findings adjusted for the baseline difference in BV. I'm not sure what the authors' response about this means - that BV at baseline is included in the baseline stream.

Reviewer #3: Mehta and colleagues' revised manuscript and response to reviewers sufficiently addressed my comments from my prior review, with one minor exception (first comment below). I also include a few comments related to minor typos I saw in the text that are important for the science and can be fixed quickly.

Minor comments

Thank you for your response to my prior comment about baseline BV being different between the arms. Even though you adjust for baseline BV, please revise the text (line 289 in clean, 296 in track changes) to not say that baseline BV was similar/reflect the difference between arms. 

Line 211 in clean, 213 in track changes - "Quality and primer trimmed" should read "Quality filtered and primer trimmed" or something along those lines.

Make sure to reference Table 4 in Adjusted, Sub-group, and Sensitivity Analyses subsection of results.

The crude BV OR in table 3 is 0.76, but in the text it's 0.74. Please correct as needed.

Signed: Kayla A. Carter

[LINK]

---

## [Decision Letter · Decision Letter 3]

26 May 2023

Dear Dr. Mehta,

Thank you very much for re-submitting your manuscript "The effect of menstrual cups on the vaginal microbiome, Bacterial vaginosis, and sexually transmitted infections: results of a nested analysis within a cluster randomized controlled trial" (PMEDICINE-D-22-03601R3) for consideration at PLOS Medicine.

I have discussed the paper with editorial colleagues and our academic editor, and it was also seen again by one reviewer. I am pleased to tell you that, provided the remaining editorial and production issues are fully dealt with, we expect to be able to accept the paper for publication in the journal.

[LINK]

Please let me know if you have any questions, and we look forward to receiving the revised manuscript.

Sincerely,

Richard Turner PhD

Consulting Editor, PLOS Medicine

plosmedicine@plos.org

Requests from Editors:

We ask you to amend the title to the following: "Analysis of bacterial vaginosis, the vaginal microbiome and sexually transmitted infections following provision of menstrual cups in Kenyan schools: Findings of a nested study within a cluster-randomized controlled trial".

Please move the trial registration number (line 17) to the end of the abstract. 

Please remove the information on study funding from the title page (in the event of publication this will appear in the article metadata, via entries in the submission form). 

At line 45, please make that "It is not known ..." or similar (reserving "We" for the authors). At line 48, that could become "We assessed ...".

At line 58, please soften the language, and we suggest "In this study, we found that menstrual cups were ...".

At line 59, we ask you to amend the text to "Further research should investigate the constitution of the vaginal microbiome and incidence of bacterial vaginosis in adolescent girls using menstrual cups".

At line 104, please refer to the CONSORT attachment by name here (see below). 

At line 428, please adapt the text to "... the study was not powered ...". 

At line 430, please soften the language to "We believe that our findings generalize to adolescent ...".

At line 438, in a request from our academic editor, please amend the sentence beginning "These results generated ..." to "These results generated from a randomized study design suggest there are beneficial impacts of menstrual cups on the composition of the vaginal microbiome and a reduction in BV. "

Noting the description at line 445, the attached CONSORT checklist will need to be labelled "S1_CONSORT_Checklist" or similar and referred to as an individual supplementary file. 

Please adapt the header for figure 1 to "Participant flow diagram" or similar. 

Please convert journal names to non-italic text in the reference list. 

Noting reference 1, please use the journal name abbreviation "PLoS ONE".

Noting reference 29, please list up to 6 author names, followed by "et al.".

Comments from Reviewers:

*** Reviewer #2: 

In my first two reviews, I restricted my comments to the most serious problems with the study design. I did not try to make a comprehensive listing of things that would need to be changed for publication, as I wasn't recommending a resubmission. In its current form, I still do not think the paper should be published.

The authors clearly want this to be the seminal paper that shows that menstrual cups reduce BV and increase Lactobacillus concentrations. This is evident from their statements at the end of the abstract, in the author summary, and the conclusion. For example, the conclusion states, "These results generated from a randomized study design provide strong evidence for beneficial effects of menstrual cups on the composition of the vaginal microbiome and reduction of BV." I appreciate that the authors have added the statement, in the limitations, that the results may not be generalizable because screening and treatment for BV is not standard of care. However, the fact that participants were screened and treated for BV means that the conclusions, as written, are not well supported by the data. The paper is written as though it is a comparison of menstrual cups versus no menstrual cups, rather than a comparison of menstrual cups plus screening and treatment for BV vs. screening and treatment for BV.

My summary of the conclusions, looking at the same data, is that in a study of screening and treatment for BV plus menstrual cups vs. screening and treatment of BV alone, the arm receiving screening and treatment of BV plus menstrual cups had lower BV and higher Lactobacillus crispatus. As screening and treatment of BV in asymptomatic women with a vaginal Gram stain score >=7 is not recommended and would not be a realistic intervention for scale up in this setting, the findings point to the need for a trial comparing menstrual cups to no menstrual cups to determine whether the menstrual cup alone is a beneficial intervention. The finding in the paper is interesting, but the authors are over-selling it as a definitive study showing the benefit of menstrual cups - this paper doesn't do that. As it stands, the abstract, author summary, first paragraph of the discussion, and conclusion, in addition to most of the body text, obscure rather than illuminate what the study really examined, what it can teach us, and what it cannot.

Major Comments:

1) On lines 207-208, and again on lines 433-434, the authors explain that girls with a Nugent score of 7-10 were treated due to lack of correlation of reported symptoms with BV. The statement that symptoms don't correlate with Gram stain score is true, but irrelevant, since treatment in this situation is not recommended. This should be removed.

2) Throughout the paper, including abstract, author statement, and body text, the authors should make it clear what the actual intervention and control conditions were, rather than trying to obscure this and use a single statement in the limitations. Understanding this study design issue is critical to understanding the results - and the need for a definitive trial of cup vs. no cup.

***

[LINK]

---

## [Editor Report · Decision Letter 4]

7 Jun 2023

Dear Dr Mehta, 

On behalf of my colleagues and the Academic Editor, Dr Stock, I am pleased to inform you that we have agreed to publish your manuscript "Analysis of Bacterial vaginosis, the vaginal microbiome, and sexually transmitted infections following the provision of menstrual cups in Kenyan schools: results of a nested study within a cluster randomized controlled trial" (PMEDICINE-D-22-03601R4) in PLOS Medicine.

PRESS

Sincerely, 

Richard Turner PhD

Consulting Editor, PLOS Medicine

plosmedicine@plos.org